# Effective Music Teachers and Effective Music Teaching Today: A Systematic Review

**Eleonora Concina**

Department of Philosophy, Sociology, Pedagogy, and Applied Psychology, University of Padova, 35122 Padova, PD, Italy; eleonora.concina@unipd.it

**Abstract:** (1) Background: This systematic review focuses on identifying the main features of effective music teachers and teaching recently examined in the educational and psychological literature. It aims to identify how recent studies have discussed the promotion of effectiveness in the context of both preservice and in-service music teachers. (2) Methods: A search in the main scientific databases for educational research (Eric, Science Direct, WWS, Web of Science, JSTOR) was conducted using keywords associated with the topics of effective teachers and teaching in the field of music instruction. In the end, thirty-six papers were identified and analyzed. (3) Results: The main themes were related to various dimensions of music teaching and teachers: teachers' personal characteristics (self-esteem, resilience, etc.) and personality traits; professional skills; cognitive and psychological aspects of teachers' professional identity (self-efficacy, professional motivation, beliefs regarding teaching and learning music, etc.); training experiences (pre- and in-service); social competence and the interpersonal relationship between the teacher and the students. (4) Conclusions: These dimensions seem interrelated and contribute to simultaneously define the effective music teacher and effective teaching in music. Specific attention should be paid to the impact of learning contexts on teachers' activities, leading to a contextualized definition of effective music teachers.

**Keywords:** music education; music teaching; effective teacher; teachers' training; systematic review

## 1. Introduction

What is necessary to be an effective teacher in music education and instruction today? This question is fundamental not only to determine the main features that characterize the figure of the effective music teacher and the main elements of effective music teaching, but also to develop relevant hints and indications that can allow us to improve music teachers' initial training and professional development. Exploring the concept of an effective teacher in music is a complex task since this construct involves several interrelated dimensions. In addition, different beliefs regarding the meaning of "effectiveness" in music education, the teacher's role in music instruction, and the teacher's impact on students' learning may make it more difficult to examine the core features that could define the figure of the effective music teacher [1]. To encourage a systematic approach to educational research pertaining to this theme, some main concepts should be clarified. These definitions should also take into account the characteristics of the current sociocultural situation, such as the fact that the world is constantly and rapidly changing, as well as the educational approaches, aims, learning goals and practices associated with music education [2] and with all other disciplines. Accordingly, understanding the core characteristics of effective music teachers is a process that could support the development of training experiences for teachers, with the final goal of allowing them to acquire the competences necessary to enhance and support students' learning processes in many different learning contexts [3].

Before discussing the main themes of the analysis, it is necessary to make a distinction. In the current survey, the topics of effective music teachers and effective music teaching are viewed as connected topics. More specifically, when considering the multidimensional

nature of the figure of the effective teacher, effective teaching can be considered to constitute one of the core components of the profile of the music teacher, which also includes relational, motivational, reflective, and emotional dimensions.

Interest in examining the main characteristics of effective teaching has increased since the end of the 1940s [4]. Being defined as a "good" or "excellent" music teacher is a process that depends on several elements. Depending on the type of learning objectives in question, music educators may adopt different approaches and strategies to effectively reach the planned educational goals. Teachers must also adapt to specific learning contexts, which are characterized by different sociocultural and economic backgrounds. In addition, teachers must adapt to each student, considering their personal characteristics, attitudes, objectives, needs, potentialities, etc. The role of students has gained increasing relevance in the teaching–learning process; therefore, pupils' learning outcomes and feedback are currently considered to be indicators of teaching effectiveness [5].

Recently, music education has shifted from a pedagogical paradigm based on the transmission of traditional praxis to an approach that is more focused on the overall development and personal growth of pupils. If music is considered a human mode of aesthetic communication that involves creative and emotional dimensions [6], the main educational goals of music education should shift from "learning to play" to "learning to live with music". The development of a personal and critical aesthetic taste, the enhancement of creative skills, the acquisition of transferable skills, and the empowerment of personal well-being are all aspects that should be considered in the context of music education and instruction. This situation implies the need to rethink learning objectives and restructure teaching methods and strategies. The role of the teacher is evolving from "master" to "tutor" and "learning facilitator" [7], and the need to change and revise some teaching approaches and strategies is becoming urgent. Teachers should support students in becoming autonomous [8] with respect to managing the process of music learning. Accordingly, what are the characteristics of effective teachers in music education at present? What has changed since the past? These questions have a great deal of theoretical and practical relevance since identifying the main construct associated with this theme may offer relevant hints for training music teachers and improving the educational experience for music students.

Considering the current situation and the need to provide hints and suggestions that can empower teachers' competences in music education, a systematic review of the literature has been conducted. The aim of this review was to examine the research concerning the topic of effective music teachers and teaching, and to define the main features of the figure of the effective music teacher and the effective process of music teaching. Educational implications for both teachers and students are also considered and included in the discussion.

## 2. Materials and Methods

Three main research questions have guided the current investigation:

(1) What are the main features related to effective music teachers and effective music teaching that have been examined recently in educational research?
(2) How has the research viewed the analysis and promotion of effectiveness in both preservice and in-service music teachers?
(3) What is the role of teacher training (preservice and in-service) in enhancing the characteristics of effective teaching in music education according to the research?

### 2.1. Inclusion Criteria

Some criteria for selecting papers have been defined:

(1) The articles must present a research study (qualitative or quantitative) pertaining to the theme of effective music teachers and teaching.
(2) The articles must have been published between 2002 and 20221 in a peer-reviewed journal in the field of educational and psychological research.
(3) The articles must be written in English.

(4) The terms "effective/effectiveness" and "music teacher/teaching" must be included in the keywords or the main topics of the articles.
(5) Participants must be pre- or in-service specialist music teachers (not generalist teachers who also teach music in their classes).

### 2.2. Search Strategies

First, an online search was performed on the main scientific databases in the field of educational and psychological research. The keywords selected to discover possible studies pertaining to the topic of interest were "effective", "effectiveness", "teacher", "teaching", and "music", which were combined in the following ways: "effective AND teacher AND music", "effective AND teaching AND music", "teacher AND effectiveness AND music", and "teaching AND effectiveness AND music".

Second, online scientific databases were selected, specifically those that included research articles dealing with educational sciences, psychology of education, and pedagogy. The database search returned 1338 results overall. From these initial results, 94 articles were excluded due to duplicated records or for other reasons, for example, some documents were not considered for the current analysis since they were not specifically articles from peer-reviewed scientific journals (but, instead, conference papers, book chapters, etc.). The remaining articles were screened, examining their main focus, topics, and research aims; only the articles which considered the topics of effective music teacher and teaching as main research theme were taken into account. Papers which dealt partially which these topics or mentioned them as secondary theme were excluded, and as a final result, 3025 papers were identified from ERIC, 65 from Science Direct, 8 from WorldWideScience, 4 from Web Of Science, and 1 from JSTOR, for a total number of 4932 articles, among which forty-seven were available in full-text version and downloaded in full-text version.

Third, the full-text version of each paper was reviewed, and articles that did not meet the inclusion criteria were discarded. This process led to the rejection of 11 papers for the following reasons:

- Some articles were essays explicating and discussing a specific theoretical claim (n = 4).
- Other articles were reviews of the literature (rather than empirical studies; n = 2).
- Others were articles offering practical suggestions for music teachers (n = 2).
- One article was a research study that addressed effective teaching from a general perspective (thus lacking a specific focus on music education).
- One article was the report of a validation process for an assessing instrument.
- One article was mainly focused on the effectiveness of a training course for preservice music teachers.

Figure 1 summarizes the search procedure in accordance with the PRISMA statement [9].

### 2.3. Analysis of the Articles

The remaining 36 articles were reviewed with respect to some specific features, which were entered into a table on an Excel sheet. For each included study, certain specific aspects were examined, i.e., year of publication, participant characteristics (nationality, professional profile, number of participants), type of research methodology adopted, main variables/factors considered in the research, and main findings. The results of this analysis are reported in Table 1.

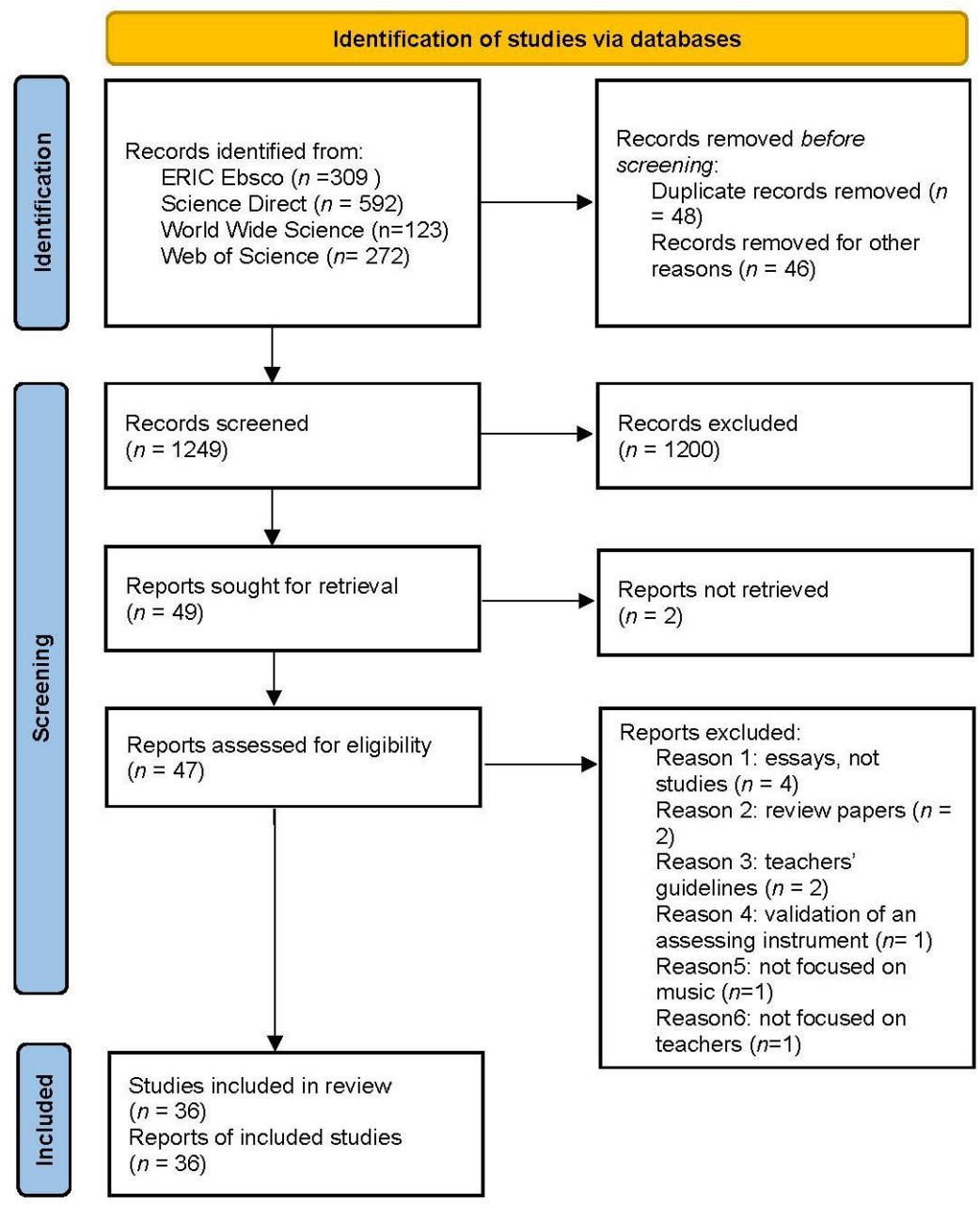

**Figure 1.** Flow diagram (based on PRISMA statement) summarizing the search procedure.

**Table 1.** Systematic analysis of the documents considering authors, years of publication, nationality of the research, type of document, main topic and main findings for each document.

| Author(S) | Year | Participants | Research Methodology | Factors/Variables Considered | Results |
|---|---|---|---|---|---|
| Ayçiçek, B. [10] | 2021 | Turkish music preservice teachers (university students, n = 28) | Qualitative methodology (case study) | Preservice teachers' beliefs about the role of critical thinking and training about this topic. | The possibility to attend a course on critical thinking helps students to develop critical thinking skills, to consider different point of views, and to reflect upon their academic and professional experiences. |

**Table 1.** *Cont.*

| Author(S) | Year | Participants | Research Methodology | Factors/Variables Considered | Results |
|---|---|---|---|---|---|
| Baker, V. D. [11] | 2012 | American music educators (Texas) who worked in all the school grades of urban discricts (elementary, middle, and high school), n = 158 | Mixed-method (questionnaire as instrument, descriptive statistics for participants' responses, analysis of open-ended questions) | Perceptions of urban music teachers' characteristics (effective vs ineffective); future career plans; urban/suburban students' characteristics, effective teachers' factors of personality; challenges in teaching and methods for assessing success in music teaching | One of the main challenges (especially for novice teachers): classroom management. The possibility to encourage and support changes in students' life as one of the main positive outcomes referred. Personal traits that sustained a long-term effective professional experience were passion for teaching music, beliefs of positive influence on students' life, realistic educational goals, and psychological resilience. |
| Ballantyne, J. [12] | 2007 | Australian novice music teachers (from 1 to 4 years of professional expertise), n = 76 respondents for the quantitative questionnaire, among them n = 15 were administered a more specific survey | Mixed-method (stage 1: questionnaire survey with quantitative descriptive analyses and means comparisons; stage 2: interviews with selected participants, qualitative analysis) | Teachers' evaluation of effectiveness of preservice training courses; impact of educational and early-years professional experiences on the perceptions of music teaching effectiveness; educational needs emerged during preservice stage. | Three main themes have emerged for the impact of training course on effective teaching: contextualization of contents and skills (considering the reality of the music class); integration (of theory and practice and of general education and music education principles); continuity (from the training to the first professional experiences in the class). |
| Ballantyne, J., Zhukov, K. [13] | 2017 | Australian novice music teachers (n = 14) | Qualitative survey (as a part of a wider mixed method research project): interviews and typological analysis of data | Professional identity of music teachers, motivation to become a music teacher, mentoring experiences, possible gaps between expectations and reality, self-efficacy in teaching music. | Music teachers' professional identity includes aspects related to positive emotions (passion for music and professional satisfaction), engagement (effort for developing professionally and for encouraging students' love for music), relationships (positive interactions with colleagues, students, and families), meaning (encouraging love for music as a main aim of music education), and achievements |

**Table 1.** *Cont.*

| Author(S) | Year | Participants | Research Methodology | Factors/Variables Considered | Results |
|---|---|---|---|---|---|
| Biasutti, M., Concina, E. [14] | 2018 | Italian vocal and instrumental music teachers (n = 160) | Quantitative survey (questionnaire for data collection, correlation and regression analyses). | Music teachers' professional self-efficacy, beliefs about learning and music ability, social skills. | Professional self-efficacy of music teachers seems to be influenced by the expression of some social behaviors (negative influence of assertiveness and expression of negative feelings), by the presence of beliefs of musical ability as incremental (positive influence) and of some personal and demographics characteristics, as gender and professional expertise (with women and more expert teachers expressing a higher level of self-efficacy). |
| Biasutti, M., Concina, E., Deloughry C., Frate, S., Konkol, G., Mangiacotti, A., Rotar Pance, B., & Vidulin, S. [15] | 2021 | Music teachers from several European countries (Croatia, Ireland, Italy, Poland, Slovenia, UK, and others, n = 335) | Quantitative survey with a set of close-ended questionnaires (correlation and regression analysis as main statistical analysis | Professional self-efficacy of music teachers, motivation to professional activity, job satisfaction, resilience, coping strategies, and self-esteem. | A high level of teachers' professional self-efficacy can be predicted by intrinsic motivation toward professional activity, resilience, coping strategies based on planning, and high level of self-esteem. Conversely, using coping strategies which are based on passive acceptance of stressful events may negatively affect the development of self-efficacy in teaching. |
| Biasutti, M., Frate, S., Concina, E. [16] | 2019 | Italian vocal and instrumental in-service music teachers (n = 24) | Mixed research method (a quantitative close-ended questionnaire, and focus group and interview). | Impact of a blended training course for in-service music teachers on their professional development, professional expertise and teaching effectiveness. | The course seemed to be effective in promoting student-centered approach in music education, effective practices and teaching strategies |
| Blackwell, J. [17] | 2020 | US expert instrumental music teachers (n = 2) | Qualitative methodology | Features of one-to-one relationship in music lessons which are typical of effective teaching. | One of the most frequent teacher's behavior is side coaching; minor errors did not result in a stop of student's performance, in order to give them the most relevant information maintaining high their motivation to learn. |

**Table 1.** *Cont.*

| Author(S) | Year | Participants | Research Methodology | Factors/Variables Considered | Results |
|---|---|---|---|---|---|
| Burton, S L. [18] | 2011 | American and Swedish preservice music teachers | Qualitative study, based on a case study considering an international collaborative training course for preservice music teachers | Impact of the course on students' academic experience, interpersonal and intercultural exchange and development of attitudes towards new strategies for music education. | The course proposed managed in promoting cultural awareness and a pedagogical attitude towards learner-centered approach among music students. |
| Button, S. [19] | 2010 | English music teachers (n = 26) | Quantitative method (a close-end questionnaire made specifically for the study, with Likert-scale responses; statistical analysis with factorial analysis) | Teachers' perceptions of the main aspects of the effective music teachers | The factorial analysis of the questionnaire revealed four kinds of teaching strategies, with references to the main approach music teachers showed in their music classes: pupil-orientated; evaluative orientated; management orientated; subject-orientated. |
| Daniel, R., Parkes, K. [20] | 2015 | Instrumental music teachers (n = 171) from nine different countries (Finland, South Africa, US, Denmark, New Zealand, Sweden, Norway, England, and Australia) | Qualitative research method (interviews and content analysis of the responses) | Participants' experience as music students: teacher–students relationships, learning cycle during first lessons, music learning in tertiary education, most influential teachers | The main teaching strategy is based on one-to-one tuition, with the support of other learning tasks (listening to expert performance, music ensemble lessons, and self-regulation in learning). Students considered as indicators of teaching expertise teacher's pedagogical and performance skills: other relevant factors in an effective teachers are some personality traits, motivation and enthusiasm toward teaching music, and social skills. |
| Howard, S. A., Seaver, K. J. [21] | 2013 | US novice music educators (n = 9) | Quantitative research method (a close-ended questionnaire) | Role of teacher's interpersonal skills in music teaching: skills that can enhance music lessons and skills that could impair them. | After a training course on professional development, participants showed an enhancement of the social skills they considered crucial for teaching music classes. One of the main strategies used to encourage changes in this dimension is self-reflection upon the social and interpersonal component of their professional activity. |

**Table 1.** *Cont.*

| Author(S) | Year | Participants | Research Methodology | Factors/Variables Considered | Results |
|---|---|---|---|---|---|
| Johnson, C., Williams, L., Parisi, J., Brunkan, M. [22] | 2016 | For the first stage of the study: an American expert teacher of swimming. For the second stage: an American instrumental music teacher | Case study with a qualitative methodology (analysis of videorecorded lessons and application of the most effective features to a musical setting) | Main features of an effective teaching cycle (instruction, behavior and feed-back/reinforcement) | Applying the elements emerged in effective swimming lessons to music lessons, they were very effective in promoting a significative and quick learning process. Clear goals, focus on the procedures, immediate feedback and reinforcement and ignoring negative behaviors seem to be key point for an effective music lesson. |
| Juchniewicz, J. [23] | 2010 | American music teachers (n = 40), 20 "exemplary teachers" and 20 "challenged teachers", and expert music educators and preservice music teachers as external evaluators (n = 84) | Mixed method: participants filled in an open-ended questionnaire examining social intelligence in terms of correctness of understanding of social situations, and their teaching practices were rate by a panel of experts. | Social intelligence and its role in enhancing effective music teaching. | No significant differences emerged between "exemplary" and "challenged" teachers, meaning a similar level of social intelligence in the two groups. Expert raters considered social skills as one of the main aspects that can influenced their evaluation of teacher's effectiveness. |
| Krause, A. E., Davidson, J. W. [24] | 2018 | American, European, and Australian expert music educators (with more than 20 years of experience, n = 12) | Qualitative methodology (open ended interviews) | Practices for music learning which could emphasize the effort for a life-long learning music learning process. | Five main categories related to the role of music education and music educators in supporting a long-lasting effort in music learning have emerged. They are Understanding and using a range of music; Finding authentic voices; Strong, sensitive and attuned to provisional knowledge; Activating opportunity in learning; Engendering commitment. Effective music educators may promote all these aspects by encouraging in students the development of competence, relatedness, and autonomy in learning. |
| Kupers, E., van Dijk, M., van Geert, P. [25] | 2015 | A Dutch expert music teacher and four students | quantitative methodology | Variations of teaching strategies in relation to the student in one-to-one tuition | When the teacher interacted with high-performing students, contingent scaffolding is more frequently used, and there was more variability in interpersonal interactions. |

**Table 1.** *Cont.*

| Author(S) | Year | Participants | Research Methodology | Factors/Variables Considered | Results |
|---|---|---|---|---|---|
| Kurkul, W. W. [26] | 2007 | American music college teachers (n = 60) and their non-music major students | Quantitative method (an audio-visual test for measuring non-verbal sensitivity, a close-ended questionnaire for assessing effectiveness of the music lesson) | Relationships between music teacher's non-verbal behaviors in music classes and the effectiveness of the lesson. | Non-verbal sensitivity of teachers, more than non-verbal behaviors, may positively influence the evaluation of music teaching effectiveness from students and external observers. |
| Legette, R. M., Royo, J. [27] | 2021 | US music educators majors (n = 4) | Multiple case study with qualitative methodology | Influence of peer-feedback about music teaching on preservice music teachers effectiveness | Feedback received from the peers was considered more empathetic than those received from the professor. Receiving comments and hints from the colleagues was very useful for improving teaching competences and feeling socially supported. |
| Leijen, A., Linde, R., Kivestu, T. [28] | 2015 | Estonian violin teachers (n = 58) | Quantitative research method (a close-ended questionnaire) | Teachers' perceptions of their professional identity and reported teaching activity. | At the beginning of their professional career teachers seem to be more focused on the subject taught (subject-centered approach), then, while acquiring expertise, they shift to a more student-centered educational approach. |
| López-Íñiguez, G., Pozo, J. I. [29] | 2016 | An expert Finnish cello teacher and a 7-year old student. | Case study with qualitative analysis of observative data (using the system for analyzing the practice of instrumental lessons) | Teacher's conceptions about teaching and learning and the instructional practices she uses during the music lessons. | The teacher, in general, adopted a constructive approach, centered on the student. In some cases, she shifted to a more teacher-centered approach, with younger students who needed to be supported in the management of some learning and cognitive processes. In general, the conceptions of the teacher are coherent with her actions during the lessons. |

**Table 1.** *Cont.*

| Author(S) | Year | Participants | Research Methodology | Factors/Variables Considered | Results |
|---|---|---|---|---|---|
| Madsen, K., & Cassidy, J. W. [30] | 2005 | US undergraduates and graduates music students (n = 78: 26 of junior level without teaching experience, 26 junior and senior level with some teaching experience, 26 graduates with full time experience in music teaching). | Mixed method: participants viewed the video-recordings of some music lessons and they have to rate the effectiveness of the teachers and the learning performance of students (quantitatively) and to comment both teacher's and students' behavior in class (qualitatively). | Perception of student teachers of other music teachers' effectiveness in terms of instruction, delivery and classroom management, and students' learning in terms of social and academic skills. | Role of expertise in assessing music teachers effectiveness: teachers with more expertise are more critical while evaluating their colleagues. Student teachers tended to focus more on teachers' behavior than on students' behavior while assessing teaching effectiveness. |
| McLeod, R. B. [31] | 2018 | US undergraduate students (music and non-music), music teachers and external observers (respectively, n = 23, n = 2, n = 4) | Qualitative methodology: open end questions to the participants for assessing the perceived effectiveness of the instructional modes presented. | Effectiveness of three instructional modes in music instruction: non-verbal, co-verbal, and verbal | Five dimensions of effective music lessons emerged; they are related to specific instructions and feedback, delivery skills and eye contact, audible and focused co-verbal instruction prompts, conducting effectiveness, and ensemble progress. The verbal mode was considered the most effective, since it included feedback from the teacher. |
| Millican, J. S., Forrester S. H. [32] | 2019 | US in-service K-12 music teachers (n = 898) | Mixed-method: a questionnaire with responses on Likert scale and open questions for assessing he relevance of teaching practices. | Definition of effective teaching practices in music according to the perception of music teachers. | One of the most relevant dimension in effective teaching is represented by the teacher–student relationship and the teacher's sensitivity to students' needs and problems. The teachers' training is also important, specifically in the preservice phase. |
| Mills, J. [33] | 2002 | English college music students (n = 182) | Qualitative method (questionnaire with open-end questions) | Students' perception of the characteristics of effective music instruc-tion(instrumental and vocal) | For both instrumental and vocal instruction, three effective teaching styles emerged: transmission, collaboration, and induction. |

**Table 1.** *Cont.*

| Author(S) | Year | Participants | Research Methodology | Factors/Variables Considered | Results |
|---|---|---|---|---|---|
| Napoles, J., McLeod, R. B. [34] | 2013 | US preservice music teachers (n = 75) | Quantitative methodology: questions asking to rate on a Likert scale teacher's effectiveness and students' behavior in video recorded lessons. | Preservice teachers' perception of teaching effectiveness considering four conditions (high delivery-high student's progress, high delivery-low student's progress, low delivery-low student's progress, low delivery-high student's progress) | Music students reported that they considered the high delivery conditions more effective, irrespective of students' high or low progress in music learning. |
| Powell, S. R., Weaver, M. A., Henson, R. K. [35] | 2018 | US preservice music teachers involving brass and woodwinds teaching courses (n = 135) | Quantitative methodology: participants' teaching performance was recorded and assessed by some external judges on a quantitative rating scale. | Level of effectiveness of preservice music teachers considering skills related to instrument setup and tone production. | All the participants scored higher in teaching brass than teaching woodwinds (woodwind is probably a more varied category). Brass and woodwind players are judged more effective in teaching their main instrument than other instrumentalists teaching brasses or woodwinds. |
| Powell, S. R., Parker, E. C. [36] | 2017 | US music educators majors (n = 134) | Qualitative methodology: participants are asked to write an essay with their beliefs about effective and ineffective music teachers | Preservice teachers' ideas about effective and ineffective music teachers. | The main characteristics of effective teachers are related to interpersonal skills and knowledge-based competences. Abilities to personalized learning experiences considering the specific needs for each students are mentioned. Unsuccessful teachers have negative attitudes toward teaching, are not motivated and show no respect for students. |
| Regier, B. J. [37] | 2021 | US band directors in high schools (n = 610) | Quantitative methodology: close ended questionnaires (with responses on Likert scale) | Teaching self-efficacy and possible relations with mastery experiences, verbal persuasions, vicarious experiences, and physiological state. Possible correlations between self-efficacy and self-perceptions of effective teaching. | Self-efficacy for teaching strategies seems influenced mainly by mastery experiences; then, on a minor level, by verbal persuasion, physiological state and vicarious experience. |

**Table 1.** *Cont*.

| Author(S) | Year | Participants | Research Methodology | Factors/Variables Considered | Results |
|---|---|---|---|---|---|
| Robinson, J. A. [38] | 2022 | Australian novice music teachers (n = 59) | Qualitative methodology | Features related to motivation, stress, value of music teachers. | Motivational aspects are mainly related to students' personal and artistical growth, success in lesson planning and opportunities for performance. Considering stressors, novice teachers referred a wide set of effective coping strategies for facing them. |
| Saygi, C., Kirmizi, F. S. [39] | 2012 | Turkish university students who were studying for becoming primary or music teachers (primary teachers n = 227, music teachers n= 63) | Quantitative methodology | Preservice teachers' self-efficacy in teaching | Preservice music teachers showed a higher level of perceived efficacy than primary music teachers. |
| Schmidt, M. [40] | 2008 | A US novice music teacher | Qualitative case study | Role of mentorship relations with supervisors in enhancing teaching effectiveness in the musical field. | Three main dimensions could be identified in empowering teaching competences: the pedagogical style of the mentors, the integration of different strategies and educational model, the achievement of coherence between teaching knowledge and teaching practices. |
| Spieker, M. H. [41] | 2017 | US novice and expert music teachers (n = 16) | Mixed methodology: analysis of language and analysis of frequencies. | Use of figurative language during verbal instruction in music lessons and its efficacy. | Expert music teachers used significantly more figurative language in music lessons than novice and student teachers. |
| Stavrou, N. [42] | 2020 | Cypriot music students and teachers (respectively, n = 518, N = 71) | Mixed methodology (questionnaire with open- and close-ended questions) | Characteristics of effective music teachers | Students teachers considered dimensions related to personal traits and interpersonal aspects crucial in the teacher–student relationship; instead, in-service music teachers rated aspects related to pedagogical knowledge and teaching strategies as more important. |
| Swanwick, K. [43] | 2008 | US music classes involved in the project Youth Music (n = 10) | Case study methodology | Musical and teaching abilities that are characteristics of music leaders. | Three main dimensions of teachers defined as good or effective: interest in music as human language, interest in students' musical and personal development, attention to musical expressivity and fluency. |

**Table 1.** *Cont.*

| Author(S) | Year | Participants | Research Methodology | Factors/Variables Considered | Results |
|---|---|---|---|---|---|
| Taylor, D. M., Raadt, J. S. [44] | 2021 | US music teachers (n = 575) | Quantitative study | Evaluations of teaching effectiveness based on some stereotypical features (namely the masculinity of the voice of male teachers) | The more masculine voices are rated as being more characterized by leadership, effective management of the class and wisdom. |
| Woody, R. H., Gilbert, D., Laird, L. A. [45] | 2018 | US music preservice students (music majors, n = 110) | Quantitative study (with a close-end questionnaire for data collection) | Values attributed to four dimensions of effective music teacher related to reflectivity, empathy, musical comprehensiveness, and beliefs about musical learning and skills. | Older college students showed a great self-appraisal and valued the dimension of reflectivity, musical comprehensiveness and beliefs about musical learning more than their younger colleagues. |

## 3. Results

Two types of analyses were performed on the collected articles. First, the frequencies of some features (year of publication, nationality of the study and the participants, and research methodology) were identified to highlight the main trends in the current educational research concerning the topic of effective teachers and effective teaching in music education. Second, considering the main variables or factors examined with respect to each study, a categorization of the main dimension emphasized by the research concerning effective music teachers and teaching was made.

### 3.1. Year of Publication (Frequencies)

The main aspect that was considered in the analysis was the year of publication. The number of articles published each year seems to have increased over the past two decades (there was only a decrease in 2020, perhaps due to the general interruption of activities resulting from the COVID-19 pandemic). This finding highlights the increasing interest in the topic of effective teaching and teachers in music education. The results are reported in Figure 2.

### 3.2. Nationality of the Study and Participants (Frequencies)

The nationalities of the study and those of the participants included different countries worldwide. Most of the research studies included (N = 17, more than half) were conducted in the US, while several research projects involved international research teams and participants to facilitate comparison of the data across different cultural backgrounds (n = 4). Other nationalities mainly included participants from European countries, while a few studies involved countries outside the EU. The prevalence of American and European countries is mainly linked to the fact that the studies were focused on music education in the Western tradition. The results are reported in Figure 3.

### 3.3. Research Methodology (Frequencies)

Considering the research method adopted in the study, the articles examined employed quantitative, qualitative, and mixed methods in nearly equal numbers. This fact highlights the different aims (exploratory or confirmative) that characterized the studies examined. The frequencies of the different kinds of research methods are reported in Figure 4.

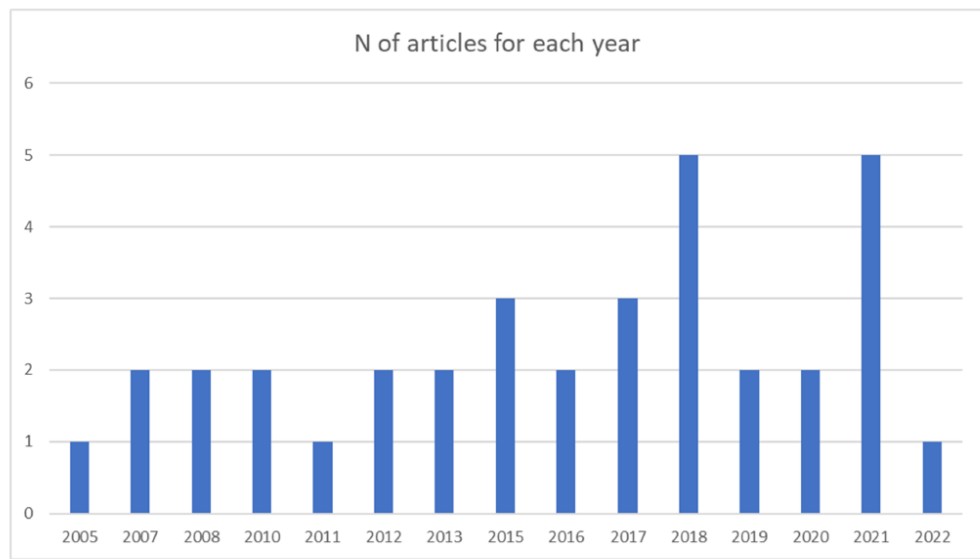

**Figure 2.** Documents published for each year (frequencies).

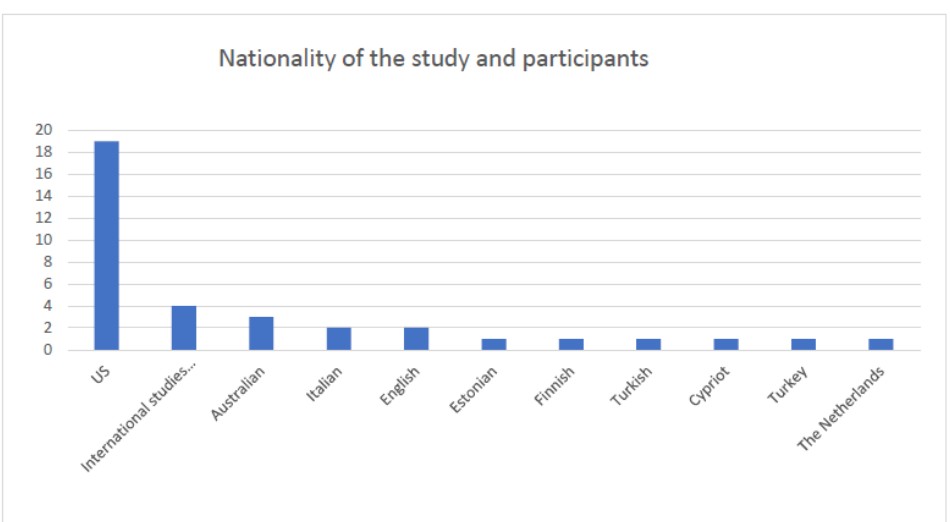

**Figure 3.** Nationality of the research (frequencies).

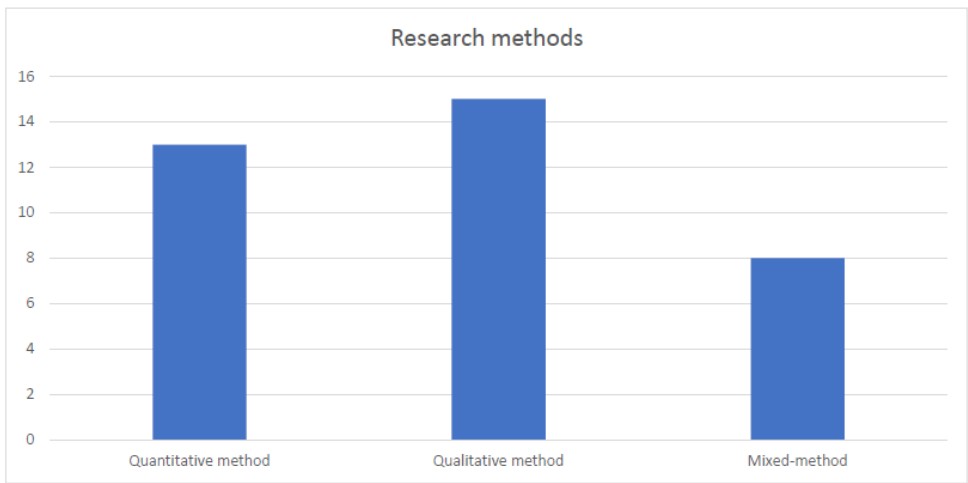

**Figure 4.** Type of research methodology (frequencies).

*3.4. Dimensions of Effective Teaching and Teachers (Categories)*

Considering the main factors and variables examined in the research included in the review, some categories emerged. The number of studies that focused on each category is reported (considering the fact that some articles involved more than one factor/variable). Factors that are mentioned only once have been reported for two general categories, one in the category of teaching and musical competences and another in the category of personal cognitive, motivational and emotional aspects.

1. Pedagogical approach to teaching and teaching strategies (n = 9);
2. Role of training experiences (preservice and in-service, n = 7);
3. Interpersonal relations and teacher's social competence (n = 5);
4. Personality traits (n = 5);
5. Professional self-efficacy (n = 4);
6. Communication and communicative style (n = 4);
7. Professional expertise and motivation (n = 3);
8. Instructional cycle (n = 2);
9. Performing skills (n = 2);
10. Other aspects related to teachers' personal, cognitive and emotional aspects (n = 7);
11. Other aspects related to teachers' professional competences (n = 4).

Each category is described in further detail below.

### 3.4.1. Pedagogical Approach to Teaching and Teaching Strategies

In music teaching, the pedagogical approach may influence the way in which the lesson is structured, as well as the meaning that is attributed to the teacher–student relationship. Music teachers may adopt different kinds of teaching strategies with respect to the main focus of the teaching approach (the student, the subject, the evaluation, or the management of the lessons, [19]). The most effective teaching approach seems to be the student-centered approach [20,28,29,32,42]; in this approach, the teacher is focused on meeting students' needs and developing innovative strategies for personalizing the teaching process [32]. In the student-centered approach, the basic assumption is that each student has unique characteristics; it is the teaching process that must adapt to the student rather than the opposite. Although this teaching approach is considered to be the most effective approach in general, it is possible, in some cases, to introduce additional teacher-centered learning tasks, especially in the context of younger students, who are less self-regulated and must be guided during the initial phases of the music-learning process [29]. Scaffolding is a teaching strategy useful with high performing students, who may also already have self-regulated skills [25] and who may benefit from the mediation of the teacher as a learning facilitator. Another strategy frequently used with music students is represented by side coaching [17], where the teacher offers relevant indications while the student is performing. In side coaching, to avoid frequent and useless interruptions of a student's performance, the instructor has to select the most urgent and relevant issues to direct the pupil's attention to them. The general pedagogical framework of the most effective teaching strategies seems to be similar both for instrumental and vocal music instruction [33], underlining that overall learning objectives may be very similar: presenting students with performing models, and supporting them in understanding, achieving, re-elaborating and consolidating these patterns.

### 3.4.2. Role of Training Experiences (Preservice and In-Service)

One feature that is frequently identified as a predictor of effective teaching is the impact of training experiences, both in terms of preservice courses in academic contexts [12,18,27,32] and in-service proposals for professional development [16,21,40]. With respect to preservice training, most relevant experiences occur in university contexts, where courses for teacher training are offered. The aspects that emerged as the most significant for supporting the development of effectiveness in teaching are related to the

integration of theory and practice and the connection between the academic world and the professional contexts in which the prospective teachers are to conduct their professional activity [12]. These courses may be the proper contexts in which to train teachers in general pedagogical knowledge and competences, as well as to allow them to develop a student-centered approach in music education [27] and the specific social and communicative skills that are necessary to manage the teacher–student relationship effectively [32]. In addition, in preservice experiences, student teachers may share ideas, proposals, and feedback with their colleagues and work collaboratively to accomplish learning tasks [18], thus laying the foundations for the creation of a professional community of practice that could support teachers in their future work. After music teachers have started their professional careers, they may benefit from a lifelong learning process of professional development. These opportunities may help participants develop effective teaching strategies and an awareness of the importance of adapting the educational process to suit students' needs [16] from the perspective of a student-centered pedagogical approach. For preservice training, in-service professional development experiences may enhance social competences to improve teacher–student relationships, which are considered to constitute a core component of effective music teaching [21]. Additionally, the importance of creating a community of practice for music teachers is particularly evident during the first phases of their professional careers, when they may benefit from advice and tutoring from more expert colleagues [40].

### 3.4.3. Interpersonal Relations and Teachers' Social Competences

As previously noted, when discussing the role of training, the social dimension plays a key role in music teaching [13,20,23,32,35]. This dimension not only refers to the interpersonal relationship between teacher and student, but also, from a broader perspective, takes into account all the social interactions that the music teacher has with his or her colleagues, institutional figures, and families [13]. An effective music teacher has a good level of social competence [23], which is necessary to develop positive relationships with students that are based on mutual respect and attention to others' needs and requests [32]. This requirement is particularly relevant since one of the most frequent teaching strategies is based on one-on-one instruction [20], which features a direct interaction between teacher and student without the mediating role of classmates.

### 3.4.4. Personality Traits

Becoming an effective music teacher may be easier if certain personality traits are present [42]. These traits may facilitate the process of developing a good level of professional competence, particularly by influencing interactions with the students and the levels of effort that they invest in their educational activity. For example, effective music teachers seem to be more resilient [11,15] charismatic and emphatic [20], particularly in the social interactions with students. In addition, they often exhibit positive attitudes toward people, music and teaching [43], a characteristic that may sustain the motivation to work as music teachers.

### 3.4.5. Professional Self-Efficacy

A personal sense of efficacy related to professional competence seems to be positively associated with teaching effectiveness in music education [46]. The development of a good level of professional self-efficacy may depend on certain aspects. First, it may be influenced by social skills, since positive interpersonal relationships in class may enhance teachers' perceptions of effectiveness [14]. Second, personal aspects such as coping strategies and resilience [15] can positively affect the development of self-efficacy among music teachers as well as their professional expertise [14]. Third, in accordance with Bandura's theory [47], previous experiences in terms of both mastery experience and vicarious experience may contribute to increasing teachers' sense of being effective in their educational activities [37]. In general, music teachers exhibit a higher level of self-efficacy than other teachers [39],

which is probably because teaching music requires specific musical skills and knowledge, which music teachers achieve by means of a lengthy and demanding training process.

### 3.4.6. Communication and Communicative Style

Communication is a core dimension of music teaching; effective teaching is characterized by a multidimensional communicative style, including eye contact, nonverbal communication, and modulation of the teacher's voice [34]. Since communication is also characterized by a musical dimension, nonverbal behaviors and sensitivity are particularly relevant to the task of transmitting indications and feedback to students [26]. Verbal instructions are also relevant in music lessons, but they must be specific and coherent with musical and nonverbal communicative hints [31], such as through the use of figurative language to help students understand complex contents [41].

### 3.4.7. Instructional Cycle

A dimension of effective teaching that is particularly associated with activity in class includes the structure of the instructional cycle. The instructional cycle refers to the sequence of phases that characterize the process of delivering a lesson [31]. The most effective cycle of instruction seems to include clear learning objectives, a focus on the procedure, quick feedback concerning student performance and potential reinforcement when required standards have been achieved [22].

### 3.4.8. Professional Expertise and Motivation

Professional experience may be a predictor of teaching effectiveness and self-efficacy [14] since, during their educational career, music teachers may experiment with different approaches and strategies, find solutions to problems, and change and renovate traditional teaching practices to comply with new pedagogical developments. Expert teachers are also more precise and knowledgeable regarding their work and their evaluations of their colleagues' activity [30]. Moreover, motivation is a relevant factor for effective music teachers [38]: teachers are motivated in their professional activity by the possibility to positively affect students' personal and artistic growth.

### 3.4.9. Performing Skills

Music teachers are also good musicians; they must have a thorough knowledge of instrumental or vocal techniques to act as a model for students [11]. Accordingly, playing music with students offers a visual and acoustic model that can be considered and followed. A good level of performing skills is crucial for integrating verbal explanations with practical examples in a process of modeling and scaffolding [35].

### 3.4.10. Other Aspects Related to Teachers' Personal, Cognitive and Emotional Aspects

Some aspects linked to psychological, cognitive and emotional constructs seem to be related to effective music teaching. Considering the cognitive dimension, critical-thinking skills may enhance the improvement of teaching practices [10]. In terms of the emotional and motivational aspects, effective music teachers are characterized by a deep personal interest in music and its meaning as a form of human expression [43], which also manifests in their motivation to teach music [35] and the positive emotions that are associated with musical activity [13]. Another relevant aspect that could influence the way in which the teacher manages their educational relationship with the student is orientation toward music skills and learning, which is considered to be incremental and can, thus, be potentially improved via learning experiences [45]. In addition, it has to be mentioned that there may be some aspects related to personal appearance that may influence the evaluation of teacher's effectiveness, for example, the masculinity of their voice [44]; it is important to be aware of these stereotypical beliefs in order to avoid incorrect judgements about the effectiveness of music teachers.

### 3.4.11. Other Aspects Related to Teachers' Professional Competences

In the professional profile of a music teacher, certain other elements may serve as conditions of effective teaching. The studies examined in the current analysis have highlighted the role of classroom management abilities [11], especially for teachers who manage collective music classes (ensembles, choirs, bands, etc.). In addition to musical competences (performing skills), the teacher must also possess a good level of general pedagogical knowledge [42] to be able to adopt the most effective strategies for introducing and presenting musical contents and teaching performing practices. They must also adopt a reflective attitude toward the educational activity, with the aim of monitoring and improving their activity both in and outside the class [45]. From an innovative perspective, the music teacher should become a learning facilitator for her or his students, encouraging them to gradually become autonomous and self-regulated with respect to their learning process [24].

## 4. Discussion

The current survey systematically examined the most significant literature concerning the topics of effective teachers and effective teaching in music instruction. The analysis considered research studies published over the past 20 years. The aspects that emerged during the analysis can be divided into two main groups: features that can impact and enhance the effectiveness of music teachers and music teaching, and the characteristics of effective music teachers. These aspects are summarized in the model proposed in Figure 5.

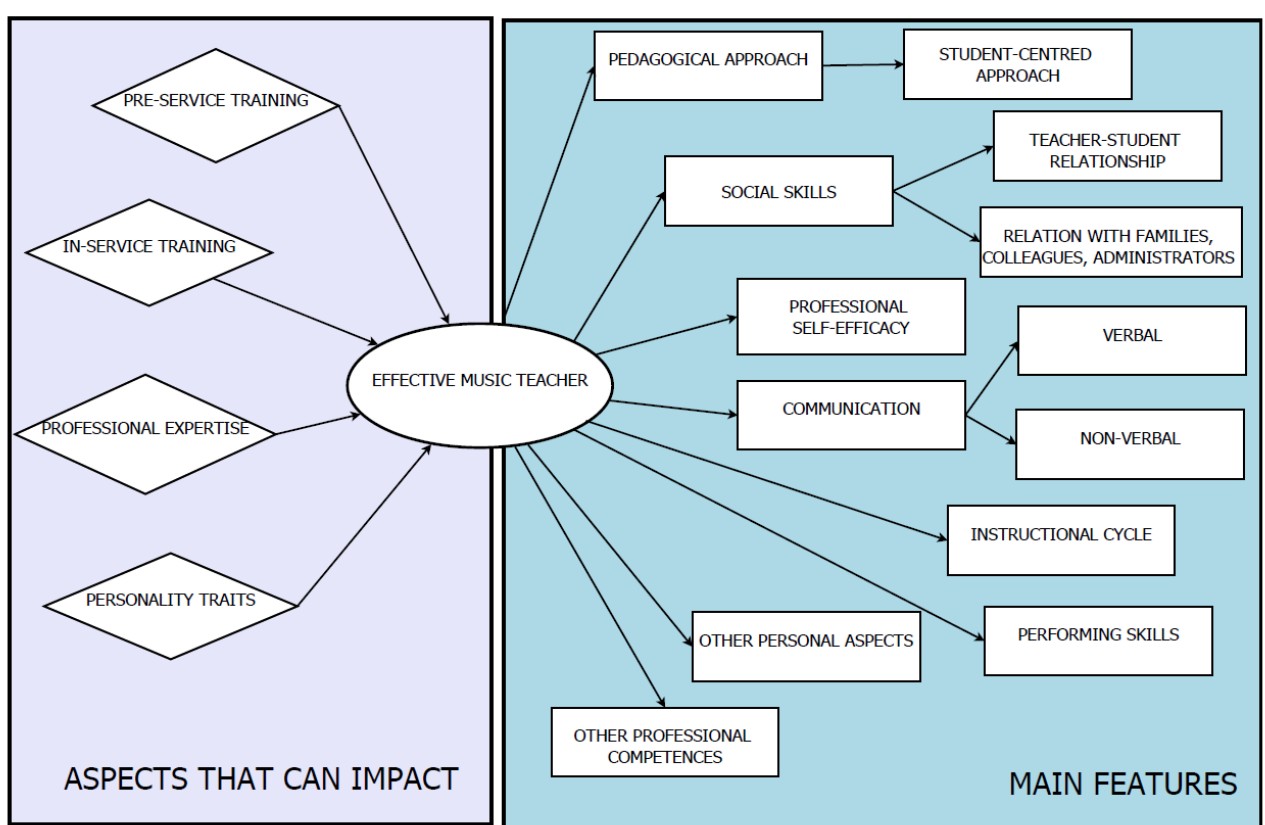

**Figure 5.** Model of the effective music teachers.

Considering the aspects that may influence the development of effectiveness in music teaching, teacher training plays a crucial role since, in this context, prospective teachers can discover and employ innovative teaching strategies and relevant pedagogical contents. Particular attention should be given to preservice pedagogical experiences because these experiences may contribute to increasing teachers' awareness of the role of a student-centered approach in music lessons as well as the importance of personalizing students'

learning experiences. Too frequently, these features are underestimated in music teachers' preservice courses: according to Allsup [2], in many cases, preservice training remains rooted in a vision of teacher–student relationships that is based on the master–apprentice model. This situation continues to be the case because it represents the most suitable approach if the main goal of learning is the transmission of traditional praxis without a focus on personal re-elaboration and reconstruction. The need to promote the full development of pupils, not only in a musical context but also holistically, implies the need to train music teachers to act as learning facilitators [24] with the aim of gradually supporting students in becoming autonomous learners.

Considering the main characteristics of the profile of effective music teachers, the current model is coherent with the categories identified by the literature review conducted by Brand [1], who identified two main dimensions of effective music teachers: the first such dimension is related to personal features (personality traits, intelligence, attitudes, psychological aspects, etc.), while the second is focused on the professional competences that are needed for activities in class (class management, teaching strategies, etc.). The characteristics associated with the nonverbal component of communication and self-efficacy are reported by Steele [48], who added to these characteristics the notion of servant leadership, a dimension that has not appeared as a relevant research theme in the current analysis. All these aspects are closely connected with interpersonal relationships with students, which can be considered to constitute one of the core dimensions of effective music teachers.

The professional competences of effective music teachers are related to two main areas, i.e., the musical field (e.g., performing skills) and the pedagogical field (pedagogical knowledge, teaching methods, etc.). This component is what Townsend [7] defined as "artistry" in the context of the profile of the effective music teacher. Music teachers are "artists" in the sense of being "masters", not only with respect to their expertise in musical techniques and performing but also with regard to their pedagogical and educational knowledge.

Comparing the promotion of teaching effectiveness in preservice and in-service music teachers, the main themes investigated are linked to the development of a student-centered approach in music lessons [27] and the achievement of the social and communicative abilities that are needed for an effective interpersonal relationship. For in-service teachers, professional needs are mainly related to promoting teachers' awareness of the need to personalize music lessons to take into account the specific characteristics of the pupils [16].

In relation to the role of in-service and preservice training experiences for promoting effectiveness in music teaching, there are many aspects that should be addressed in training courses. A topic which should be addressed during the very first years of university is related to the teaching strategies and how to use them effectively in accordance with the educational context and students' needs and characteristics [17,25]. Training experience should connect the theoretical dimension of music education and instruction and the real world of music instruction [12], preparing prospective music teachers to face the multidimensional contexts of music classes. Theoretical concepts should be supported by the promotion of relational and communicative skills [32], which are fundamental for structuring a positive and fruitful relation with the students. Another transversal aspect which is crucial for music teacher's professional development is the enhancement of critical-thinking skills [10], as a fundamental competence for reflecting, monitoring and evaluating individual educational activity, in order to improve both teachers' and students' educational experience.

The current analysis faces certain limitations. First, all the studies examined focused on music education and instruction in the Western tradition. No information was provided concerning music teaching in cultural backgrounds outside the Western context. It would also be useful to examine the characteristics of effective teachers with reference to music teaching in other cultures in order to identify peculiarities and similarities from an intercultural perspective. Second, according to the search criteria, some potentially relevant papers were excluded; for example, no papers that were related to academic and educational

conferences were considered, although many relevant international conferences focus on themes related to music education and instruction. Finally, most of the studies examined refer to instrumental music teachers or do not make a specific distinction in their analysis between vocal and instrumental teachers. This aspect does not allow one to make specifical distinctions between effective features in instrumental and vocal music, highlighting peculiar aspects that are typical of each of these two teaching–learning processes.

## 5. Conclusions

The current systematic review aimed to define the complex profile of effective music teachers. Considering the results that emerged, the figure of the effective music teacher is characterized by several dimensions, which are mainly related to (1) aspects that can support the development of effectiveness in teaching, (2) components of the effective teacher that pertain to personal characteristics, and (3) components of the effective teacher that are related to professional competences (musical and performing skills, general pedagogical contents, and teaching strategies). According to this general model of effective teachers, there is an overarching characteristic that can be identified with flexibility, including cognitive, behavioral, and relational flexibility. This characteristic is the main feature that allows a teacher to adapt successfully to the socio-educational context, to the institutional environment, and, more relevantly, to each pupil's characteristics and needs. To adapt in a flexible manner, it is necessary to critically reflect on one's own personal activity in order to adjust and regulate educational and teaching activities to promote a contextualized learning experience. Flexibility and critical thinking are, from this perspective, some of the most relevant transferable skills that should be included in music teachers' training to help participants become "effective music teachers".

**Funding:** This research received no external funding.

**Institutional Review Board Statement:** Not applicable.

**Informed Consent Statement:** Not applicable.

**Data Availability Statement:** Not applicable.

**Conflicts of Interest:** The author declares no conflict of interest.

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
