# Peer review of "Effective Music Teachers and Effective Music Teaching Today: A Systematic Review"

_education, doi:10.3390/educsci13020107_

Round 1

Reviewer 1 Report

line 113 - What other reasons were articles excluded?

line 117 - "and aAs"

24 in table - A Dutch expert
music teachers
and four students  

 Incorrect grammar

25 in table - (a audio-
visual test for
measuring non
verbal sensitivity

incorrect grammar

28 in table - maybe depending
upon the young age of the
students,

incorrect grammar, also do not use "maybe"

29 in table – juunior

31in table - Mixed-mehod:

          teacher-students

32 in table – thre

40 in table – Us should be US

43 – stereotiphical

         vioce   

44 – mayors

         skills (

great-self-appraisal

 incorrect grammar and typos

Difficult to read this table with centered format – huge spaces between words makes it hard to read

p17 - Alt-
hough (upper third of page) Should be al-though

competences (last line on page) should be competencies

Why “tuition” used throughout rather than “instruction?”

P19 - instrumental (or vocal) techniques

Why is “vocal” in parenthesis?

P20 top line - student is orientation toward music skills

It has also to be mentioned – same paragraph as above

, for avoidingdo you mean “to avoid?” in same paragraph

P20 3.4.11 - conducting performing practices. This is confusing

Line 244 – “since the first year of university” should be “during the first year ….”

Line 245 – “in accord to theshould be “in accordance…” or “in accord with the”

Line 248 – “prospect” should be “prospective”

Line 252 – “music teachers” to “music teacher’s”

Author Response

I would like the reviewer for all the suggestions and indications reported.

I have answered all the comments below.

line 113 - What other reasons were articles excluded?

Thank you for your question. I have added the following sentence, in order to clarify this:

“for example, some documents were not considered for the current analysis since they were not specifically articles from peer-reviewed scientific journals (but, instead, conference papers, book chapters, etc.)”

line 117 - "and aAs"

Thank you, I have corrected the typing mistake.

24 in table - A Dutch expert

music teachers

and four students 

 Incorrect grammar

Thank you for indicating this, I have corrected it, the correct version is “A Dutch expert music teacher and four students

25 in table - (a audio-

visual test for

measuring non

verbal sensitivity

incorrect grammar

Thank you, I have corrected it into “an audio-visual test for measuring non- verbal sensitivity

28 in table - maybe depending

upon the young age of the

students,

incorrect grammar, also do not use "maybe"

Thank you for the suggestion, I have changed it into “with the younger students

29 in table – juunior

Thank you, I have corrected the typing mistake.

31in table - Mixed-mehod:

          teacher-students

Thank you, I have corrected the typing mistakes

32 in table – thre

Thank you, I have corrected the typing mistake.

40 in table – Us should be US

Thank you, I have corrected the typing mistake.

43 – stereotiphical

         vioce  

Thank you, I have corrected the typing mistakes.

44 – mayors

Thank you, I have corrected the typing mistakes.

         skills (

Thank you, I have corrected the typing mistakes.

great-self-appraisal

Thank you, I have corrected the typing mistakes.

 incorrect grammar and typos

Thank you, I have corrected the typing mistakes.

Difficult to read this table with centered format – huge spaces between words makes it hard to read

Thank you for this comment, I have formatted the text aligning everything to the left.

p17 - Alt-

hough (upper third of page) Should be al-though

Thank you, I have corrected the typing mistakes.

competences (last line on page) should be competencies

Thank you, I have corrected the typing mistakes.

Why “tuition” used throughout rather than “instruction?”

Thank you for the suggestion, I have replaced the term “tuition” with “instruction”

P19 - instrumental (or vocal) techniques

Why is “vocal” in parenthesis?

Thank you for the comment, I have removed the brackets.

P20 top line - student is orientation toward music skills

Thanks to the comment, I have corrected the mistake

It has also to be mentioned – same paragraph as above

Thanks to the comment, I have corrected the mistake

, for avoiding – do you mean “to avoid?” in same paragraph

Thank you for the comment, I have replace it with “to avoid”

P20 3.4.11 - conducting performing practices. This is confusing

Thank for the comment, I have replaced “conducting” with “teaching”

Line 244 – “since the first year of university” should be “during the first year ….”

Thank you for the comment, I have corrected it.

Line 245 – “in accord to the” should be “in accordance…” or “in accord with the”

Thank you for the comment, I have corrected it.

Line 248 – “prospect” should be “prospective”

Thank you for the comment, I have corrected it.

Line 252 – “music teachers” to “music teacher’s”

Thank you for the comment, I have corrected it.

I have also made some corrections to the references, since one was missing in the references section, and another was not indicated as required by journal guidelines.

Reviewer 2 Report

The systematic review of the titled article Effective music teachers and effective music teaching today: A systematic review, carried out for the Journal Education Sciences (ISSN 2227-7102). Manuscript ID education-2133159

The article is a systemtic review . It carries out an important bibliographical review of the articles published in recent years on the subject under study: the effectiveness of music teaching and music teachers. It is a very new and little researched topic. It facilitates the study for further research on music teaching as it compiles a significant number of articles already published.

A set of 36 articles is analysed in great detail. Inclusion and exclusion criteria are clearly specified for all the material located after several searches in different databases and repositories. The tables and figures are relevant, explanatory and very appropriate. They show the data correctly, are easy to understand and read, and help to understand the work carried out. The bibliography is up to date and responds to the object of study.

Author Response

Thank you for reviewing my manuscript, I am glad you have appreciated my work.